# A dynamic and expandable digital 3D-atlas maker for monitoring the temporal changes in tissue growth during hindbrain morphogenesis

**Matthias Blanc[1], Giovanni Dalmasso[2], Frederic Udina[3], Cristina Pujades[1]***

[1]Department of Medicine and Life Sciences, Universitat Pompeu Fabra, Barcelona, Spain; [2]European Molecular Biology Laboratory, Barcelona, Spain; [3]Department of Economics and Business, Universitat Pompeu Fabra, Barcelona, Spain

**Abstract** Reconstruction of prototypic three-dimensional (3D) atlases at the scale of whole tissues or organs requires specific methods to be developed. We have established a digital 3D-atlas maker (DAMAKER) and built a digital 3D-atlas to monitor the changes in the growth of the neuronal differentiation domain in the zebrafish hindbrain upon time. DAMAKER integrates spatial and temporal data of cell populations, neuronal differentiation and brain morphogenesis, through *in vivo* imaging techniques paired with image analyses and segmentation tools. First, we generated a 3D-reference from several imaged hindbrains and segmented them using a trainable tool; these were aligned using rigid registration, revealing distribution of neuronal differentiation growth patterns along the axes. Second, we quantified the dynamic growth of the neuronal differentiation domain by *in vivo* neuronal birthdating experiments. We generated digital neuronal birthdating 3D-maps and revealed that the temporal order of neuronal differentiation prefigured the spatial distribution of neurons in the tissue, with an inner-outer differentiation gradient. Last, we applied it to specific differentiated neuronal populations such as glutamatergic and GABAergic neurons, as proof-of-concept that the digital birthdating 3D-maps could be used as a proxy to infer neuronal birthdate. As this protocol uses open-access tools and algorithms, it can be shared for standardized, accessible, tissue-wide cell population atlas construction.

*For correspondence:
cristina.pujades@upf.edu

**Competing interest:** The authors declare that no competing interests exist.

## Editor's evaluation

This methodological manuscript is of interest to the fields of neural development, tissue morphogenesis, and image analysis technologies. The authors developed an image registration tool and created a digital atlas to reflect the anatomical distribution of neuronal birthdates in the developing zebrafish hindbrain. The provided resources can be very useful to monitor temporal changes in tissue growth.

## Introduction

Generating a precise temporal cartography of cell populations in the three-dimensional (3D) embryonic brain is essential for understanding how brain morphogenesis impacts the position, and therefore function, of neuronal progenitors and neuronal clusters in normal and pathological conditions. Pioneer work in *Drosophila* provided 3D-registration atlases and software that can automatically find the corresponding landmarks in a subject brain, and map them to the coordinate system of a target brain (*Heckscher et al., 2014*; *Peng et al., 2011*). However, up to now, the incorporation of time

**eLife digest** The brain, like most other organs, is formed by the coordinated growth of a few unspecialized cells in the embryo, which give rise to billions of neurons. For the brain to work properly, it is crucial that, during embryonic development, each neuron ends up in the correct location. This migration to the right spot has to happen while the brain grows and changes shape, which affects how and how far neurons and their precursor cells need to move to reach their final position. If these movements and changes in shape are not coordinated correctly, neurons can end up in the wrong place, form the wrong connections, and ultimately impact how the brain works.

Previous work done in fruit flies and zebrafish resulted in three-dimensional maps of these animals' healthy brains, which allowed scientists to have a holistic view of how brains are organized. Although these maps are a valuable resource to study the structure of the brain, they do not provide information on how the brain transforms over time, especially during embryonic development. To get a clearer picture of how a few precursor cells give rise to the incredibly complex tissue that is the brain, a three-dimensional map spanning the entire developmental process is needed.

To fill this gap in knowledge, Blanc et al. developed a digital atlas-maker pipeline (DAMAKER) that allows scientists to generate three-dimensional models of the embryonic brain from microscopy images of several individuals. They then used this pipeline to construct a three-dimensional digital atlas of how a part of the brain called the hindbrain develops in the zebrafish embryo.

First, they collected images of the hindbrain showing neurons born at different times and matched these images to the existing static maps. Next, DAMAKER was used to follow neurons from the time of their birth to their final location, allowing Blanc et al. to create a map showing where neurons born at different stages during development end up. This type of map allows users to accurately determine when different populations of mature neurons are born, which allows scientists to estimate when different defects in brain development might originate.

Based on these data, Blanc et al. concluded that in zebrafish most of the cells that will end up forming the hindbrain acquire their specialized neuronal identities very early in development, between 24 and 48 hours post fertilization. These temporal maps of healthy hindbrains were then compared to maps of brains in which the birth of neurons was disrupted, thus changing the final number of neurons in the brain. This experiment showed that changing the number of neurons that are born early in development alters the final positions of neurons and the overall shape of the brain. Therefore, for the brain to grow to its correct size, there must be a balance between the number of unspecialized cells in the developing brain, and the rate at which these cells become neurons.

The DAMAKER pipeline not only provides scientists with a tool to study neurodevelopmental disorders, but also serves as a method that can be adjusted to map growth and shaping of other organs.

---

—and therefore information of tissue morphogenesis— in 3D-atlases is still one of the main challenges that needs to be solved.

Digital 3D-atlases enable the exploration of brain structures in the 3D-context and can be updated to incorporate new information, which is essential for the exponential increase of knowledge. Existing gene expression 3D-atlases in larval and adult fish provide valuable tools (*Chow et al., 2020*; *Jaggard et al., 2020*; *Kenney et al., 2021*; *Kunst et al., 2019*; *Ronneberger et al., 2012*; *Tabor et al., 2019*). However, to incorporate the temporal component has been challenging, most probably because whole brain images are particularly hard to register across space and time, even though there are recent efforts aiming to make this more accessible (*Dsilva et al., 2015*; *Fernandez and Moisy, 2020*). Temporal digital 3D-atlases are essential to understand how the generation of brain cell diversity occurs concomitantly with brain morphogenesis, which results in a dramatic transformation from a simple tubular structure (e.g. the neural tube) to a highly convoluted structure (e.g. the brain). Thus, our goal was to provide an atlas allowing (i) the incorporation of time to enable morphogenesis studies, (ii) the quantification of 3D-patterns with a standardized method, (iii) the comparison of tissue shapes, and (iv) an easy access to the atlas data with user-friendly upgradability, without the need to send the data to the atlas-makers for processing. Such features of a digital atlas are crucial for studies that rely on 3D-topologies to understand how specific regions develop; moreover, to do this in zebrafish would enhance the use of zebrafish as human avatars in

neurodevelopmental disorders by enabling to understand how specific territories respond to injury or gene disruptions.

Here, we developed a new 'Digital Atlas-MAKER' pipeline (DAMAKER) that fulfills the previous requirements and can be used to create an expandable atlas of virtually any tissue in any organism. Further, it is able to perform data analysis and quantification mostly on a single platform, FIJI, which is already established as the standard for image analysis in the field. The expandable zebrafish digital 3D-atlas was created by combining *in vivo* imaging data from distinct transgenic embryos with the Fijiyama registration tool (*Fernandez and Moisy, 2020*), which allows multi-modal rigid registration, and the Trainable Weka Segmentation (TWS) (*Arganda-Carreras et al., 2017*) that leverages a limited number of manual annotations and permits data to be automatically segmented after a short training and to quantify tissue volumes. We applied DAMAKER to the monitoring of the growth of the neuronal differentiation domain in the context of the whole zebrafish hindbrain —the embryonic brainstem—, to study how cell differentiation and morphogenesis might be intertwined. We made use of the hindbrain, which is the most conserved vertebrate brain vesicle (*Murakami et al., 2005*) and undergoes tissue segmentation along the anteroposterior (AP) axis leading to the formation of seven rhombomeres (*Fraser et al., 1990*; *Jimenez-Guri et al., 2010*; *Kiecker and Lumsden, 2005*; *Krumlauf and Wilkinson, 2021*; *Pujades, 2020*). This morphogenetic process is accompanied with the initial neuronal differentiation, which starts prior 24hours post-fertilization (hpf) with a pronounced increase from 30hpf onwards (*Voltes et al., 2019*). Neuronal differentiation involves changes in cell proliferative capacity and an extraordinary displacement of neurons from their birth site, and occurs while other dramatic changes take place, such as the generation of the brain ventricle (*Belzunce et al., 2020*; *Gutzman et al., 2008*; *Hevia et al., 2022*; *Lyons et al., 2003*). Although previous work evoked the importance of the temporal order of neuronal differentiation in ascribing final neuronal position within specific circuits (*Kinkhabwala et al., 2011*; *Pujala and Koyama, 2019*; *Wan et al., 2019*), little is known about its impact in the final organization of differentiated neurons within the hindbrain.

Here, we registered experimental whole zebrafish hindbrain images and mapped neuronal differentiation patterns with a unified spatial representation. We build up a digital 3D-hindbrain atlas in which signals corresponding to specific territories and differentiated neurons were merged. With these combinations, we could associate specific neuronal populations with anatomical landmarks and quantify complex volumetric data. Next, we performed a tissue wide neuronal birthdate analysis by using a method based in photoconverted fluorescent protein tracing *in vivo* (*Caron et al., 2008*), which permitted to distinguish early-born vs. late-born neuronal territories in live embryos. This allowed us to develop neuronal differentiation maps that revealed the remodeling of the neuronal progenitor and neuronal cluster domains over time. By generating these maps, we unveiled that early-born neurons were always located in the inner part of the differentiation domain and surrounded by younger neurons, generating an inner–outer gradient of older vs. younger neurons. As a proof-of-concept, we demonstrated that this temporal 3D-atlas could be used as a proxy to infer the birthdate of given neuronal populations, as well as to analyze neurogenic defects. We believe that this tool —the digital 3D-atlas maker, DAMAKER— will allow the integration of the information generated in other laboratories and will help ascribing neuronal birthdate and neuronal differentiation times upon gene disruption in zebrafish avatars for human diseases. This collaborative integration is essential for expanding our overall knowledge about how the brain is functionally organized, and therefore advancing brain research, medicine and brain-inspired information technology.

## Results

Our scientific interest was to monitor the dynamic growth of the neuronal differentiation domain in the context of the whole zebrafish hindbrain with the emphasis in how neuronal differentiation order or birthdate may prefigure cell position within the differentiation domain. For this, we developed a standardized and reliable 3D-imaging processing method —the digital 3D-atlas maker or DAMAKER— that allowed us to generate digital 3D-models, to quantify volumes and compare tissues shape, and most importantly for us to incorporate time.

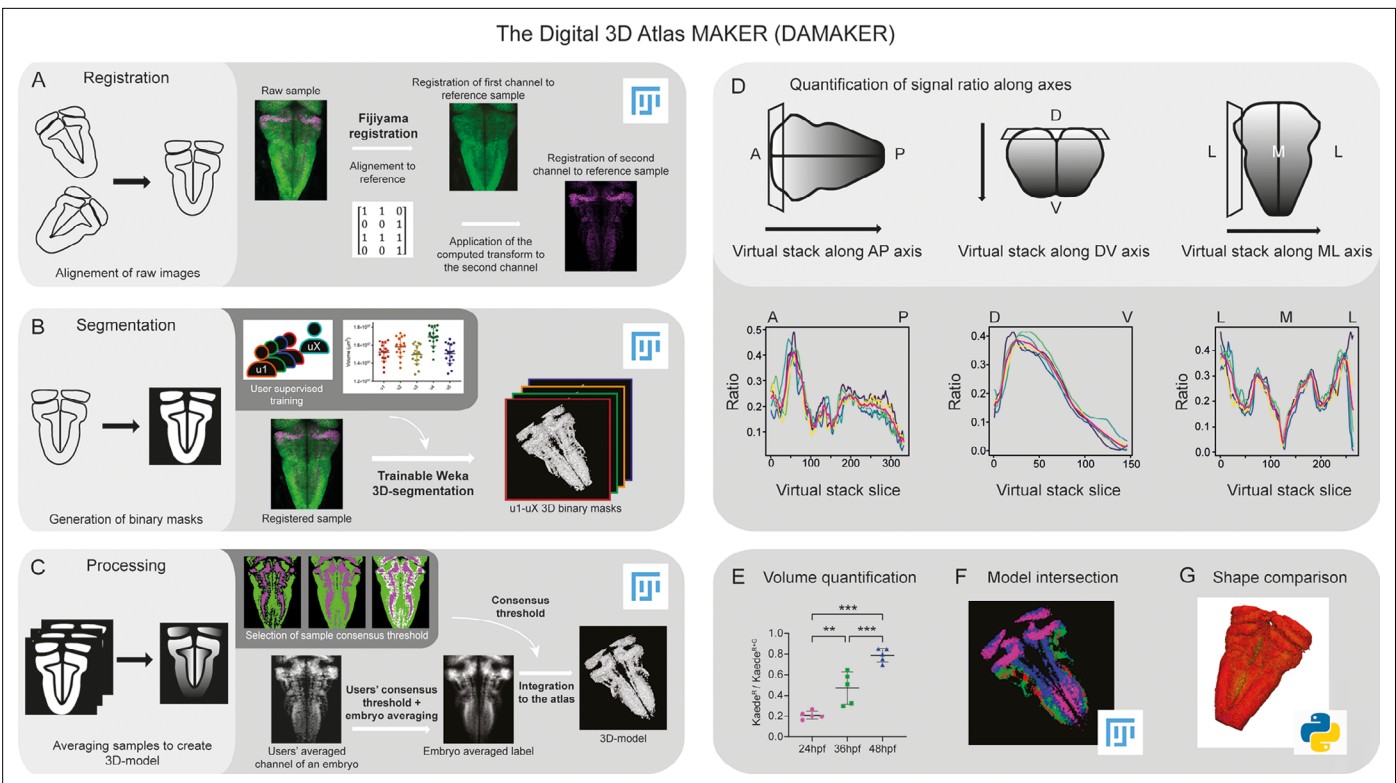

**Figure 1.** Image analysis pipeline (DAMAKER) and digital 3D-atlas construction. Schematic depiction of sample processing through the digital 3D-atlas making pipeline. (**A**) For image registration, acquired dual channel images from embryos are registered using the Fijiyama registration plugin in FIJI, the first channel (HuC:GFP signal in green) is aligned to a previously selected HuC signal which serves as reference orientation. The resulting transform file is then used to align the signal of interest (e.g. gad1b:DsRed displayed in magenta). (**B**) For image segmentation, Tg[HuC:GFP] embryos are used for supervised training of the Trainable Weka 3D-Segmentation algorithm. Images are provided to users, and each user inputs brush strokes on areas with signal and areas considered as background across the stack, in order to obtain a classifier (e.g. a set of filters and thresholds to be used by the algorithm to isolate the signal). Users' generated classifiers are compared by quantification of a dozen of HuC-signals from various embryos. User classifiers resulting in more than 10% variability among users (ANOVA test) are considered outliers (e.g. u4, user 4) and are not used for sample segmentation. For each embryo, as many 3D-binary masks as user-generated classifiers are created. (**C**) For processing, 3D-binary masks of a single embryo are averaged to obtain a users' average segmentation; this allows to assess the interuser segmentation variability and limits the human bias introduced by the supervised training. Signal similarity threshold between users can then be selected (here, consensus among all users was isolated). Users' average embryo 3D-models of a given condition (n=5) are then averaged together to obtain the embryo's averaged label. To select sample consensus threshold, we compare embryos' averaged labels at different thresholds with their negative imprint onto the neuronal differentiation domain. Signal consensus among at least three embryos seems to recapitulate best the original signal, with marginal overlap between the signal and its imprint. Sample consensus threshold is then applied to the embryos' averaged label to generate the final 3D-model, ready for the integration into the atlas. Several functions can be performed with DAMAKER such as (**D**) quantification of the signal ratio along body axes by virtually reslicing embryos' averaged models along the desired axis, (**E**) quantification of users' averaged label volumes and display them as a ratio of signal over the neuronal differentiation domain, (**F**) intersection of the 3D-model integrated in the atlas to assess overlapping or segregation of signals, and (**G**) analysis of the tissue shape.

The online version of this article includes the following figure supplement(s) for figure 1:

**Figure supplement 1.** Analyses of interindividual signal variability.

## Developing a standardized, reliable 3D-image processing and quantification method

Building upon previous work that reconstructed prototype 3D-atlases of whole embryos in *Drosophila* (*Heckscher et al., 2014*; *Peng et al., 2011*) and of larval or adult zebrafish (*Chow et al., 2020*; *Jaggard et al., 2020*; *Kenney et al., 2021*; *Kunst et al., 2019*; *Ronneberger et al., 2012*; *Tabor et al., 2019*), we aimed to create a digital 3D-atlas of the hindbrain neuronal differentiation domain that could incorporate time. To build it up, we designed a pipeline requiring the following steps: (i) registration of *in vivo* 3D-images displaying specific signals in order to align raw samples from different embryos (*Figure 1A*), (ii) segmentation of the 3D-images to generate labeling masks, each corresponding to a given marker or cell population (*Figure 1B*), and (iii) processing these masks to

create average digital 3D-models (*Figure 1C*). In addition, the pipeline should allow to isolate specific signals from *z*-stack images in a semi-automatic manner in order to (i) quantify them along any body axes (*Figure 1D*) or calculate signal volumes upon time (Figure E), (ii) generate a digital 3D-atlas by intersection of several reference 3D-models (*Figure 1F*), and (iii) perform tissue shape comparisons (*Figure 1G*).

Since we were interested in neuronal differentiation patterns, to establish a reference tissue signal we made use of Tg[HuC:GFP] embryos at 72hpf, a stage in which most of the cells have undergone neuronal differentiation and GFP is uniformly displayed in the whole neuronal differentiation domain (*Hevia et al., 2022*). Acquired 3D-images from different Tg[HuC:GFP] embryos (e1–eY; n=5each) were registered to the reference by Fijiyama registration, in which rigid registration was favored, although non-rigid registration could be considered at the cost of sample shape variability. Then, the obtained transformation matrix could be applied to other signals, such as those labeling specific tissue domains, cell populations, etc (*Figure 1A*). After alignment of 3D-images, tissue segmentation was applied in order to generate binary masks for each labeling/staining (*Figure 1B*). Here, segmentation settings were at tissue scale, both in an effort to keep computing power low for pipeline accessibility and for our interest in cell populations, although single cell resolution could be achieved with nuclear reporter lines. In order to discern noise vs. signal in the raw data from each embryo, different users (u1–uX; n=5) were invited to employ the Trainable Weka Segmentation (TWS) tool to train classifiers (one per user) (*Figure 1B*). In the case of low or sparse signals (or specific structures within a signal) specific training of the algorithm with different settings would be better to leverage the structural recognition capabilities of TWS. To reduce the potential impact of the human bias from the trained classifiers, we averaged the resulting segmented images from multiple users' trained classifiers (*Figure 1B*). In the case of having an outlier user, this classifier was not considered (as example, see u4 in *Figure 1B* as an outlier that was discarded). Accordingly, we generated as many binary 3D-masks as users (n=4; *Figure 1B*). The users' averaged embryo 3D-masks displayed a gradient, which represented the spatial similarity between users' segmentations. We selected the signals agreed by at least three users (*Figure 1B*), although this could be differently thresholded to display signal consensus among users, with the strength of the threshold setting up the diversity. For image processing, these embryo 3D-masks were averaged to merge information from all embryos (e1–eY; n=5) to obtain the 'label 3D-model'. As many label 3D-models as desired signals were generated, by reiteration of the whole process (*Figure 1C*). Each digital label 3D-model displayed a gradient, which represented the spatial similarity between embryos. To establish the most accurate consensus selection, we compared label 3D-models thresholded for different levels of consensus between embryos. We selected the signals displayed by at least 3 embryos, compared the distribution of the labeling between embryos, and built up the digital reference 3D-atlas (*Figure 1C*). To evaluate the accuracy of the registration, we assessed (i) the interindividual variability of different signals (*Figure 1—figure supplement 1A–E,A'–E'*), (ii) the size of the neuronal differentiation territory (HuC-domain) in different transgenic backgrounds (*Figure 1—figure supplement 1F*), and (iii) quantified the interindividual variability between the HuC volume of the individuals and the HuC volume of the generated 3D-model (*Figure 1—figure supplement 1G*). Thus, this pipeline allowed us to (i) assess signal similarity between segmentations limiting the bias due to the users' subjectivity, (ii) obtain reliable 3D-tissue quantifications, (iii) produce a representative 3D-model of a label, (iv) estimate interindividual variability, and (v) build up a reference 3D-brain to be further used for spatiotemporal comparisons.

## Building a multilayered digital 3D-atlas

Next, we wanted to add multiple layers to the neuronal differentiation 3D-atlas. For this, we made use of double transgenic embryos displaying a fluorescence reporter in the neuronal differentiation domain that was used as a reference for alignment in one channel, and another fluorescence reporter of the specific cell population or territory of interest (*Figure 2*). In addition, we made use of the rhombomeres (r) 3 and 5 as anatomical landmarks to have a positional coordinate system on which to align the quantitative representation of the given signal distribution across the tissue. Thus, we imaged Tg[Mu4127;HuC:GFP] embryos, which displayed DsRed in r3 and r5 (*Distel et al., 2009*), and GFP in the neuronal differentiation domain (*Figure 2A–A'*). Upon rendering the processed 3D-model of the signal, we could observe the 3D-view of the neuronal differentiation r3 and r5 volumes (*Figure 2—video 1A*). To monitor differences in the specific signal spatial distribution provided by Mu4127 within

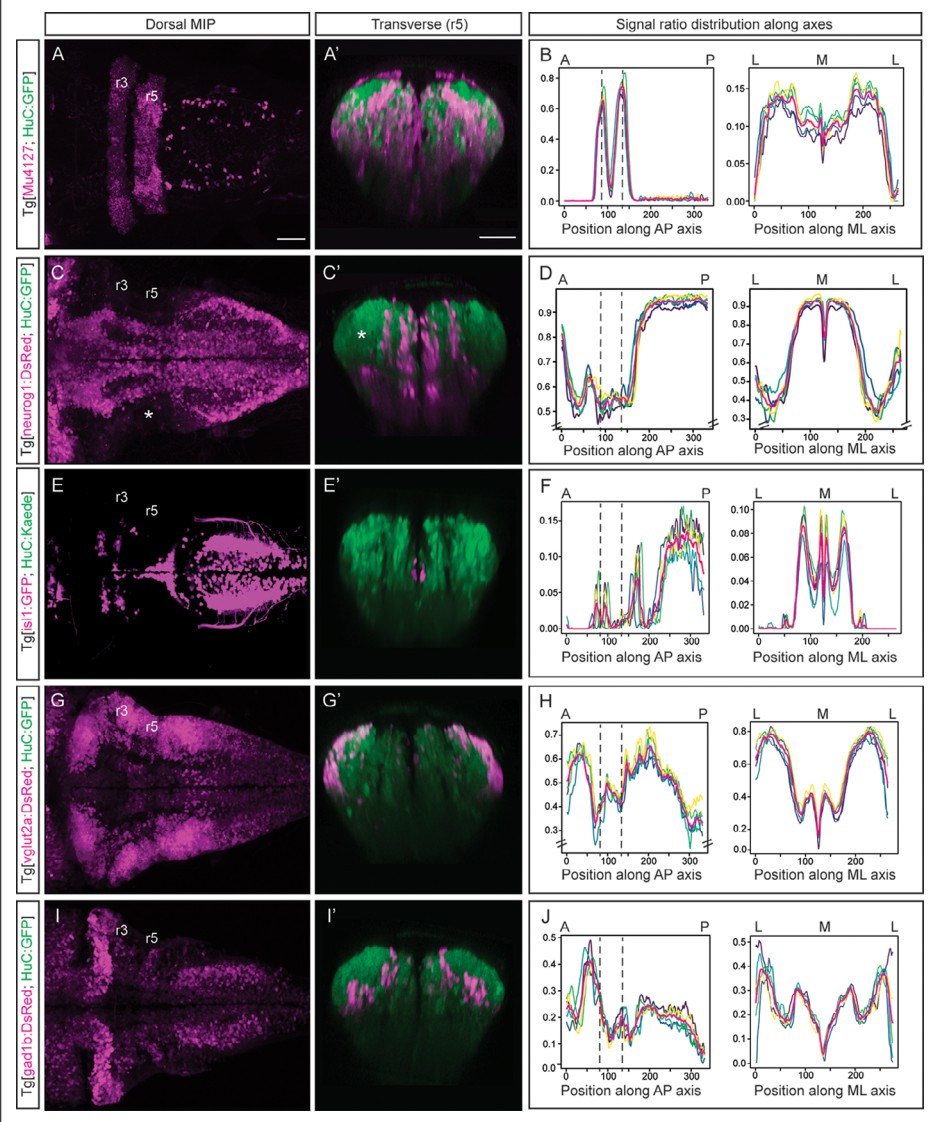

**Figure 2.** Incorporating territory landmarks allows precise spatial characterization of signals. (**A, C, E, G, I**) Dorsal maximal intensity projections (MIP) with anterior to the left, and (**A', C', E', G', I'**) transverse views through r5 of registered embryos, in the Tg[HuC:GFP] background (72hpf). (**A, C, E, G, I**) display only the magenta signals such as r3 and r5 (Mu4127), neurog1, isl1, vglut2a and gad1b, whereas (**A', C', E', G', I'**) display these signals and the HuC (green). Scale bar, 50μm. (**B, D, F, H, J**) Signal ratio distribution along the anteroposterior (AP) and mediolateral (ML) axes of different individual embryos (n=5; colored lines), with the average value displayed as magenta line. Position along the AP and ML body axes is indicated on the top of the graph. Black dashed lines parallel to the Y-axis correspond to r3 and r5 positions assessed by Mu4127 signal. Note in (**D, H**) the offset in the Y-axis for easier readability of the signal variations. Scale bar, 50μm.

The online version of this article includes the following video for figure 2:

**Figure 2—video 1.** 3D-models of rhombomeres 3 and 5, and neurog1-derivatives.
https://elifesciences.org/articles/78300/figures#fig2video1

**Figure 2—video 2.** 3D-models of different neuronal populations.
https://elifesciences.org/articles/78300/figures#fig2video2

the neuronal differentiation territory, we obtained the volumes' value across body axes by taking sample segmentations, created 'virtual slices' along axes and quantified the signal ratio at each 'slice' (*Figure 1D*; *Figure 2B*). Accordingly, the signal distribution was concentrated in r3 and r5 along the anteroposterior (AP) axis, whereas it was similarly distributed along the mediolateral (ML) axis (*Figure 2B*).

Next, we investigated the distribution of the differentiated neurons deriving from specific neuronal committed progenitors, according to their proneural gene expression (*Guillemot, 2007*), in this case *neurog1*. In Tg[neurog1:DsRed;HuC:GFP] embryos, we observed the striped-pattern distribution of *neurog1*-derivatives in the neuronal differentiated domain (*Figure 2C–C'*; *Figure 2—video 1B*). The automatic quantification of the *neurog1*-volume displayed differences in the ratio of labeling along the AP axis (*Figure 2D*), supporting the observation of neuronal differentiated regions with less *neurog1*-derivatives when compared with more posterior domains (*Figure 2C–C'*, see white asterisk in the lateral r5). In addition, the more medial distribution of *neurog1*-derivatives in the hindbrain was observed (*Figure 2D*) when compared with the spinal cord as it was previously described (*Belzunce et al., 2020*). Thus, this quantitative analysis allowed to associate signal distribution to specific territories or cell populations. For mapping distinct neuronal differentiation programs, we first analyzed the spatial distribution of motoneurons by using Tg[isl1:GFP;HuC:Kaede$^{Red}$] embryos and observed their ventral and medial position (*Figure 2E–E'*; *Figure 2—video 2A*). Upon quantification, higher signal was observed in discrete patterns along the AP axis with a clear increase in the spinal cord (*Figure 2F*). We explored the specific neurotransmitter-expressing cell populations, considering the mainly expressed markers of excitatory (glutamatergic neurons) and inhibitory populations (GABAergic neurons) in the zebrafish hindbrain (*Higashijima et al., 2004*). Using Tg[vglut2a:DsRed;HuC:GFP] and Tg[gad1b:DsRed;HuC:GFP] embryos we observed the main contribution of glutamatergic neurons to the most dorsolateral neuronal differentiation domain (*Figure 2G–G'*), whereas GABAergic neurons allocated more medially (*Figure 2I–I'*). We built up the digital 3D-maps (*Figure 2—video 2B–C*) and quantified the distribution along the axes. We observed a smaller contribution of glutamatergic neurons to the r3-r5 region (*Figure 2H*), whereas GABAergic signal at this stage was enriched in r2 (*Figure 2J*) when compared to the rest of the hindbrain. This quantification recapitulated the spatially restricted domains of neurotransmitter expression (*Figure 2G' and I'*) previously described (*Kinkhabwala et al., 2011*; *Koyama et al., 2011*; *Pujala and Koyama, 2019*). The overlay of the distinct digital 3D-maps provided information about the segregation of populations (*Figure 2—video 2D*). Overall, the expression analysis of fluorescent reporter transgenic lines allows the virtual colocalization of signals, and therefore the model intersection in order to study specific subsets of neuronal populations and their spatial relationships.

## Building the digital 3D-atlas of neuronal birthdate

Previous work demonstrated the importance of neuronal birthdate in ascribing neuronal position and function. The stripe patterning of *alx*-expressing neurons, which are involved in swimming behavior, is age-related (*Kinkhabwala et al., 2011*), and the position of the V2a cell body and the order of spinal projections in the hindbrain relies on their neuronal birth (*Pujala and Koyama, 2019*). Thus, we wanted to explore whether neuronal birthdate was a general rule that dictated the spatial distribution of differentiated neurons within the hindbrain.

To build birthdating 3D-maps of neuronal differentiation, we performed spatiotemporal analyses of neuron birthdating using Kaede photoconversion experiments as a tool to register temporality (*Caron et al., 2008*). In this case, Kaede$^{Green}$ in Tg[HuC:Kaede] embryos was *in vivo* photoconverted to Kaede$^{Red}$ at different developmental stages (24, 36 or 48hpf), and embryos were imaged either at 48hpf or 72hpf to assess the relative position of neurons born at different times within the differentiation domain (*Figure 3*; earlier-born neurons Kaede$^{Red}$-cells vs. older-born as Kaede$^{Green}$-cells). To generate the birthdate 3D-map at 48hpf, we first photoconverted the HuC:Kaede$^{Green}$ at 24hpf and imaged embryos at 48hpf. We observed that the onset of neuronal differentiation was before 24hpf and that the first differentiated neurons located more ventral and medial than late-born cells (*Figure 3A–A'*; see magenta vs. green cells in 3 A'). When we labeled the neurons born before 36hpf we appreciated that neuronal differentiation dramatically increased between 24hpf and 36hpf, since the magenta territory in this case was larger, and that later differentiated neurons piled up on the top of the previously differentiated ones (*Figure 3B–B'*; see magenta vs. green cells in 3B'). We

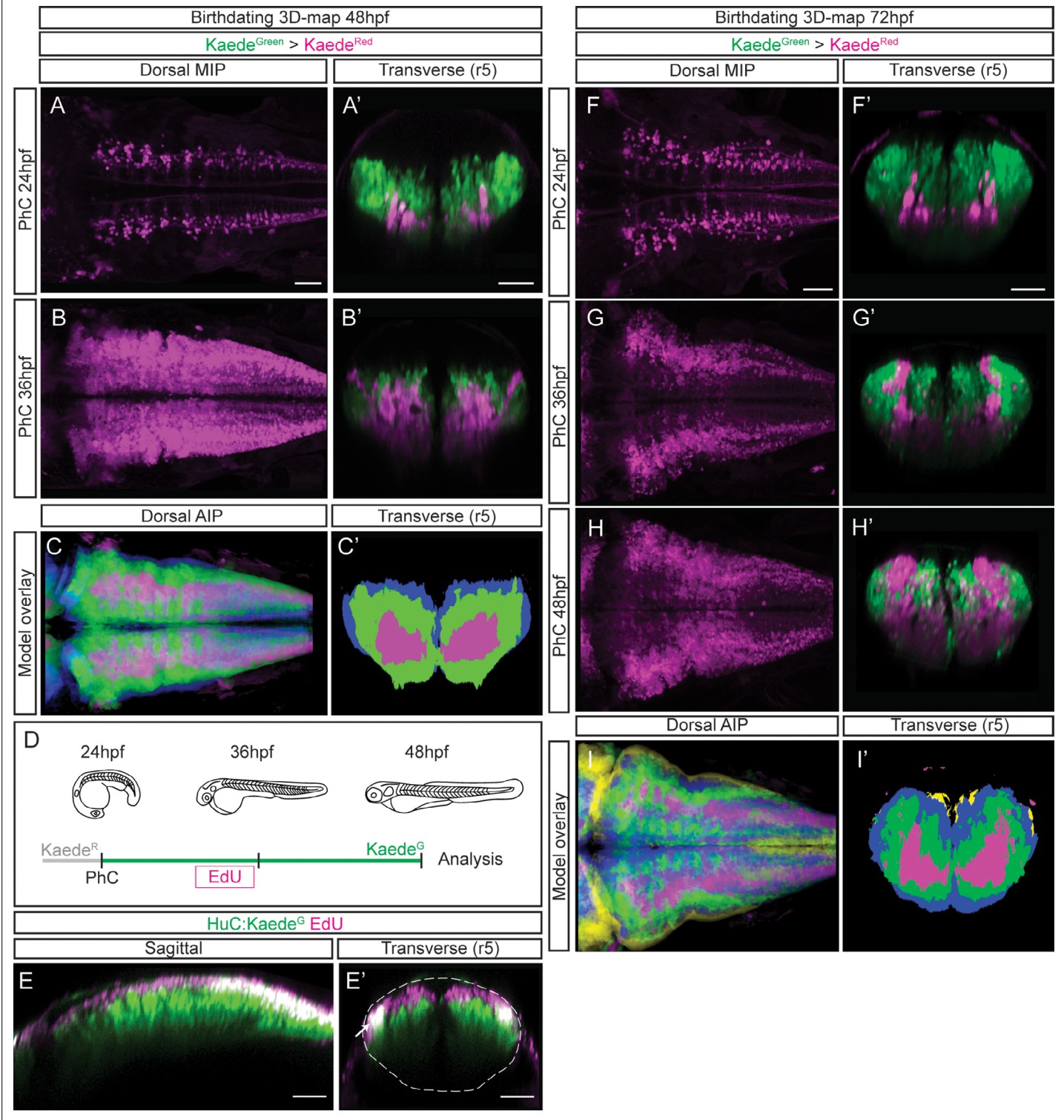

**Figure 3.** Differentiated neurons spatially organize according to their birthdate. (**A–B**) Dorsal maximal intensity projections (MIP) showing only the HuC:Kaede^Red, and (**A'–B'**) transverse views through r5 displaying both HuC:Kaede^Green and HuC:Kaede^Red at 48hpf. HuC:Kaede^Green was photoconverted at the indicated times (PhC). Note that HuC:Kaede^Red is displayed in neurons born before the time of photoconversion, and HuC:Kaede^Green in neurons born after. (**C–C'**) Neuronal birthdate 3D-map at 48hpf resulting from the intersection of photoconversion 3D-models. Neurons produced before 24hpf are depicted in magenta, those produced between 24hpf and 36hpf are in green, and the blue regions display neurons produced between 36hpf and 48hpf (n=5 embryos each). (**D**) Scheme depicting the photoconversion experiment followed by EdU-treatment. HuC:Kaede^Green was photoconverted at 24hpf, embryos were let to develop, pulsed with EdU for 6hr at 30hpf, and imaged at 48hpf. (**E–E'**) Tg[HuC:Kaede^Green] embryo displaying EdU (magenta)

*Figure 3 continued on next page*

*Figure 3 continued*

and HuC (green) in sagittal and transverse views, respectively. Note that EdU-cells are within the most dorsal part of the hindbrain, specifically in the progenitor domain and in the dorsal HuC domain (**E′**, see white arrow). The contour of the neural tube in (**E′**) is indicated with a white dashed line. (**F–H**) Dorsal maximal intensity projections (MIP) showing only the HuC:Kaede[Red], and (**F′–H′**) transverse views through r5 displaying both HuC:Kaede[Green] and HuC:Kaede[Red] at 72hpf. HuC:Kaede[Green] was photoconverted at the indicated times (PhC). (**I–I′**) Neuronal birthdate 3D-map at 72hpf resulting from the intersection of photoconversion 3D-models. Neurons produced before 24hpf are depicted in magenta, those between 24 and 36hpf in green, the ones between 36 and 48hpf in blue, and neurons produced between the last photoconversion (48hpf) and the time of acquisition (72hpf) in yellow (n=5 embryos each). Dorsal and sagittal views display anterior to the left. Scale bar, 50μm.

The online version of this article includes the following video and figure supplement(s) for figure 3:

**Figure supplement 1.** Interindividual variability analyses of the birthdating 3D-maps.

**Figure supplement 2.** Quantification of the neuronal growth volume according to cell birthdate.

**Figure 3—video 1.** The temporal registration 3D-map at 48hpf.

https://elifesciences.org/articles/78300/figures#fig3video1

**Figure 3—video 2.** The temporal registration 3D-map at 72hpf.

https://elifesciences.org/articles/78300/figures#fig3video2

further analyzed the remodeling of the neuronal differentiation territory, by generating the reference 3D-models of the early-born (24hpf) and late-born (36hpf) neurons in the context of the whole differentiated neuronal domain at 48hpf (*Figure 3—video 1A–B*; see in green neurons born between 24 and 36hpf, and in blue those born between 36 and 48hpf). We compared them to determine whether the temporal order of neuronal differentiation prefigured the spatial distribution of neurons in the tissue. We overlapped and color-coded them according to the neuronal birthdate, and observed that neurons born at different stages were precisely organized at 48hpf (*Figure 3C—C′*; see youngest neurons in blue, neurons differentiated between 24 and 36hpf in green, and the oldest ones in magenta). Many neurons differentiated between 24 and 36hpf (compare *Figure 3A′ and B′*), and they surrounded the existing differentiated neurons generating an inner-outer differentiation gradient (*Figure 3C′*). To demonstrate that indeed newly born cells were orderly incorporating into the neuronal differentiation domain, we combined HuC:Kaede[Green] photoconversion with an additional age-labeling method such as EdU-incorporation. We photoconverted HuC:Kaede[Green] at 24hpf, pulsed embryos at 30hpf with EdU to label proliferating cells, and let them grow until 48hpf (*Figure 3D*). We observed cells with EdU-incorporation exclusively in the newly differentiated domain, stacking dorsally, with no signal indicating a migration of newly born neurons in the older neuronal domain, therefore confirming the establishment of segregated domains according to neuronal differentiation timings (*Figure 3E—E′*).

To seek the impact of morphogenesis in the position of different birthdated neurons, we expanded our study to 72hpf, in which most of the cells have already differentiated. As before, we observed that early differentiated neurons were located more ventrally and medially, and that late-born cells piled up (*Figure 3F–H*, 3F′–H′; see magenta vs. green cells in 3F′–H′). To monitor possible differences in the spatiotemporal distribution within the neuronal differentiation domain, we generated the neuronal birthdate 3D-map at 72hpf and observed that the differentiation capacity diminished with time (*Figure 3I–I″*; see small yellow domain). Indeed, by 48hpf, almost the whole differentiation domain was already occupied by previously differentiated neurons (*Figure 3I–I″*; see blue, green and magenta domains). This continuous birth of differentiated neurons could be clearly observed by assessing the relative contribution of early- vs. late-born neurons to the neuronal differentiation domain (*Figure 3—video 2A–C*; see youngest neurons in yellow, those differentiated between 36hpf and 48hpf in blue, and between 24hpf and 36hpf in green). The intersection of the different neuronal differentiation intervals allowed to better appreciate the inner-outer neuronal differentiation gradient (*Figure 3I′*).

To demonstrate that the birthdating 3D-maps displayed low interindividual variability, we verified that photoconversion of the HuC:Kaede[Green] at different stages was consistent in all the analyzed embryos. For this, we first checked that the overall HuC-volumes did not dramatically change among embryos that underwent HuC:Kaede[Green] photoconversion at different stages (*Figure 3—figure supplement 1A, C*). Second, we calculated the difference between the HuC volume of the individuals and the generated models and showed that average individuals' variability from the model did not exceed 15% (*Figure 3—figure supplement 1B, D*). And third, we overlaid the HuC:Kaede[Red]

domains in 72hpf embryos that were generated at different times, and displayed them as a gray gradient (*Figure 3—figure supplement 1E–G,E'–G'*). Next, we verified that the HuC:Kaede[Red] cells labeled in an early conversion were included in the HuC:Kaede[Red] domain labeled in a late conversion. For this, we overlaid the differentiation domains contributed at each time interval (*Figure 3—figure supplement 1H–J,H'–J'*). Overall, these observations allowed us to confidently use the generated birthdating 3D-maps to estimate the importance of neuronal birthdate in predicting the neuronal position within the HuC-domain.

Next, we compared the territory occupied by neurons differentiating at distinct embryonic stages by quantifying the volume size of the neuronal differentiation domains. We detected that at 48hpf, of the whole differentiated domain (12.07 [SD ± 1.36]×10$^6$ μm$^3$), only a small part comprised neurons differentiated before 24hpf (3.42 [SD ± 1.56]×10$^6$ μm$^3$ early-born neurons vs. 8.64 [SD ± 0.86]×10$^6$ μm$^3$ late-born neurons), whereas most of the differentiated neurons arose before 36hpf (9.61 [SD ± 0.68]×10$^6$ μm$^3$) and the rest differentiated between 36 and 48hpf (3.10 [SD ± 0.35]×10$^6$ μm$^3$). This timely growth in the differentiated neuronal domain could be clearly observed when the relative contribution of early- vs- late-born neurons as compared to the whole neuronal differentiation territory at 48hpf was plotted (*Figure 3—figure supplement 2A*). When quantifying the signal ratio along the axes, we could observe similar dynamics along the AP and ML axes between 24hpf and 36hpf, and a clear difference along the DV demonstrating that neurons born at 24hpf located more ventrally than neurons born at 36hpf (*Figure 3—figure supplement 2B*). When volumes were quantified at 72hpf, we observed that only a small part of the differentiation domain came from neurons born before 24hpf (2.77 [SD ± 1.23]×10$^6$ μm$^3$ for early-born vs. 11.88 [SD ± 1.28]×10$^6$ μm$^3$ for late-born). Most neurons underwent differentiation between 24hpf and 48hpf, with half of the differentiated neurons arising before 36hpf (6.75 [SD ± 2.19]×10$^6$ μm$^3$ born before 36hpf vs. 7.58 [SD ± 2.37]×10$^6$ μm$^3$ after 36hpf) and the rest were differentiated between 36 and 48hpf (11.19 [SD ± 1.09]×10$^6$ μm$^3$ born before 48hpf vs. 2.99 [SD ± 0.90]×10$^6$ μm$^3$ after 48hpf). When the ratio of neurons born at different stages over the whole neuronal differentiation domain was assessed, a steady increase of neuronal differentiation domain was observed (*Figure 3—figure supplement 2C*; 24hpf: 0.21 ± 0.04; 36hpf: 0.47 ± 0.16; 48hpf: 0.79 ± 0.07). Upon quantifying the signal ratio along the axes, the domains corresponding to early- vs. late-born neurons followed the same trend with no main differences along AP and ML axes (*Figure 3—figure supplement 2D*; e.g., the magenta, green and blue lines did not display major changes in the multiple virtual slices). As expected, when the analysis was performed along the DV axis, differences in cell allocation relying on the neuronal birthdate could be appreciated, with early-born neurons (24-36hpf) displaying a more ventral enrichment (*Figure 3—figure supplement 1D*, see DV-axis graph).

Overall, the overlaying of birthdating 3D-maps showed that most of the neuronal differentiation events occurred between 24 and 48hpf, and early-born neurons were always located in the inner part of the differentiation domain and surrounded by younger neurons, generating an inner-outer gradient of early-born vs. late-born neurons. This suggested that the position of neurons relied on their neuronal birthdate, and revealed the the importance of morphogenesis for remodeling the differentiation domain.

## The birthdating 3D-atlas as a proxy to infer neuronal differentiation order

Our next goal was to investigate whether position in the neuronal differentiation domain at late embryonic stages could be used as a proxy for neuronal birthdating. As a proof-of-concept, we used the Tg[vglut:DsRed] and Tg[gad1b:DsRed] lines, which display DsRed in the glutamatergic and GABAergic neurons, respectively (*Satou et al., 2013*). To dissect when glutamatergic neurons were born, we generated the 3D-models of neuronal differentiation of Tg[vglut2a:DsRed] at 48 and 72hpf, and intersected them with the birthdate 3D-maps created in *Figure 3*, in such a manner that gluta-matergic neurons were color-coded according to their birthdate (*Figure 4A—A' and B—B'*). Gluta-matergic neurons contributed in a big proportion to the differentiation domain, and were born during all the analyzed temporal windows (*Figure 4—video 1A–B*). They displayed the inner-outer distri-bution depending on birth-time, with the oldest cells cells in the inner domain (see magenta cells in *Figure 4A'–B'*; *Figure 4—video 1A–B*), and the youngest neurons in the most outer position (see blue and yellow domains in *Figure 4A'–B'*; *Figure 4—video 1A–B*). In the case of GABAergic neurons, we

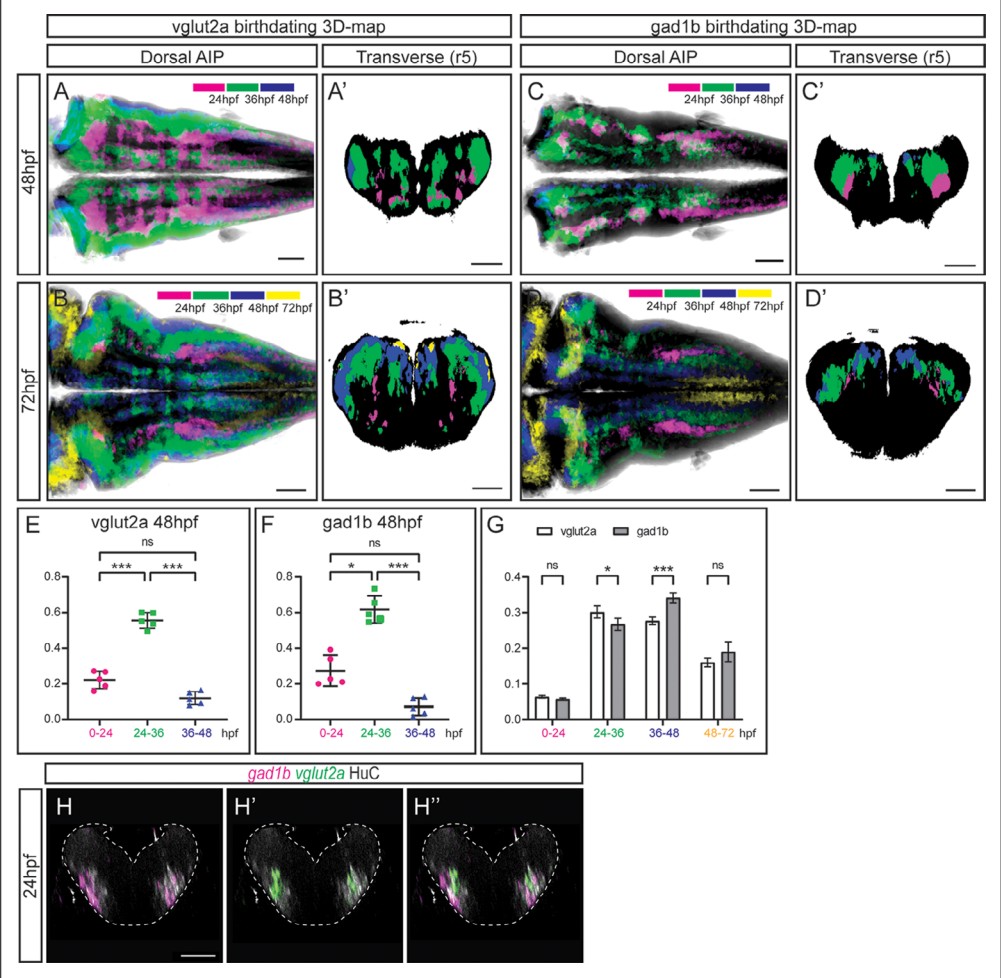

**Figure 4.** The temporal registration 3D-atlas can be used as a proxy to reveal the birthdate of the desired neuronal differentiated neurons. (**A–D**) Dorsal average intensity projections (AIP) with anterior to the left, and (**A–D'**) transverse views through r5, of the glutamatergic (**A–B**) and GABAergic (**C–D**) 3D-models intersected with the birthdating 3D-map at 48hpf (**A, C**) and at 72hpf (**B, D**). vglut2a or gad1b neurons produced before 24hpf, between 24 and 36hpf, between 36 and 48hpf and between 48 and 72hpf are color-coded as indicated. The differentiated domain is depicted in black. (**E–F**) Dot-plots showing the relative contribution to the corresponding vglut2a/gad1b-neuronal differentiation domains at 48hpf of neurons born at the indicated time intervals. (**G**) Interleaved bar-plot showing the relative contribution of glutamatergic and GABAergic neurons generated at different time intervals to the vglut2a/gad1b-neuronal differentiation domains at 72hpf, respectively. (**E–F**) RM one-way ANOVA with Tukey's multiple comparison test, and (**G**) two-way ANOVA with Šídák's multiple comparisons test; p<0.033 (*) p<0.002 (**) p<0.001 (***). (**H–H"**) Transverse views of a 24hpf Tg[HuC:GFP] embryo hybridized with *vglut2a* (green) and *gad1b* (magenta), and immunostained with HuC (gray). Images are displayed as the overlay of *gad1b* and HuC (**H**), *vglut2a* and HuC (**H'**), and the merge of the three (**H"**). Note that already at 24hpf, there are *vglut2a* and *gad1b* cells within the HuC domain. The neural tube contour is depicted with a white dashed line. Scale bar, 50µm.

The online version of this article includes the following video and figure supplement(s) for figure 4:

**Figure supplement 1.** Quantification of the signal ratio distribution of glutamatergic and GABAergic neurons along the axes.

**Figure 4—video 1.** The neuronal birthdate 3D-atlas as a proxy to reveal the order of neuronal differentiation.

https://elifesciences.org/articles/78300/figures#fig4video1

took the same approach and intersected the 3D-model of Tg[gad1b:DsRed] embryos on the birthdate 3D-map, and color-coded them according to the differentiation time. Some GABAergic neurons in r3 and r5 were born as early as 24 hpf (*Figure 4C–C'*; *Figure 4—video 1C–D*). These observations suggested that an important part of glutamatergic and GABAergic neurons was born between 24 and

36hpf, although some were born earlier and later too. Accordingly, when assessing the relative contribution of neurons born at different intervals to the corresponding neuronal populations, we observed that of the total glutamatergic and GABAergic neurons, approximately 20% and 30%, respectively, were contributed by cells born before 24hpf, 60% by cells born between 26 and 36hpf, and only by a small percentage of neurons born between 36 and 48hpf (*Figure 4E–F*). Comparison of the growth dynamics of both populations at 72hpf showed that they were slightly different, with glutamatergic neurons produced earlier than GABAergic neurons (*Figure 4G*). Overall, this analysis indicated that position of differentiated neurons at 72hpf could recapitulate neuronal birthdates.

To demonstrate that indeed this could be used as a proxy to infer birthdate, and that there were glutamatergic and GABAergic neurons at 24hpf, we performed *in situ* hybridizations with *vglut2a* and *gad1b* probes at early stages of embryonic development while monitoring the neuronal differentiation domain. Indeed, glutamatergic and GABAergic neurons were already specified at 24hpf and they were located within the HuC territory (*Figure 4H–H"*), although in our hands the GABAergic transgenic line did not display the reporter expression at this stage. Thus, these results demonstrate that the neuronal birthdating 3D-map can be used as a proxy to infer neuronal differentiation order, and widens the use of the digital 3D-atlas. To go deeper in the differentiation dynamics of the glutamatergic and GABAergic populations, we quantified the ratio of neurons differentiated in distinct time windows along the axes (*Figure 4—figure supplement 1*). Analysis at 48hpf showed that the glutamatergic populations born between 24 and 36hpf and between 36 and 48hpf underwent a drastic displacement along the dorsoventral (DV) axis, when compared to the same populations observed at 72hpf. The 24-36hpf neuronal subset, which displayed a relatively homogenous distribution along the DV axis when observed at 48hpf, was medially located at 72hpf (engulfing the 0-24hpf subset). The 36-48hpf subset, mostly dorsal when observed at 48hpf, was distributed both ventrally and dorsally when observed at 72hpf (*Figure 4—figure supplement 1A–B*). These observations support the idea that newly produced neurons initially located dorsally were displaced towards ventral, resulting in the formation of the inner to outer gradient of neuronal differentiation. In addition, the medio-lateral distribution of the neurons produced at different time intervals recapitulated the different allocations of late born glutamatergic and GABAergic neurons, with the 48-72hpf subset of glutamatergic neurons more laterally located than the GABAergic 48-72hpf subset, which was enriched medially (*Figure 4—figure supplement 1B, D*).

## Tissue shape analysis reveals defects in neurogenesis and the remodeling of the neuronal differentiation domains

To take this newly developed pipeline one step further we tested whether it could serve as a proof-of-concept to analyze aberrant neurogenesis. For this, we disturbed embryonic neurogenesis by conditionally inhibiting the Notch-pathway, which has been demonstrated to play an important role during hindbrain neurogenesis (*Hevia et al., 2022*; *Nikolaou et al., 2009*). We made use of photoconversion experiments in Tg[HuC:Kaede] embryos to discern early-born vs. late-born neurons, conditionally inhibited Notch-signaling for a short time (only affecting to newly generated neurons), and embryos were then imaged at 72hpf (*Figure 5A*). When comparing control embryos with those in which Notch-signaling was downregulated, we observed an increase in newly differentiated neurons that dramatically impacted in the early-born neuronal domain, since newer neurons occupied the most dorsal part of the hindbrain displacing the older neurons (*Figure 5B–B' and C–C'*) as previously described (*Belzunce et al., 2020*; *Hevia et al., 2022*). Accordingly, upon quantification of the neuronal differentiation domain we observed a significant increase of the HuC volume in treated embryos (*Figure 5D*), and newly produced neurons were mainly localized in the medial domain (*Figure 5E*). To deepen our knowledge on the impact of the neurogenesis increase and the remodeling of the hindbrain, we developed a protocol for tissue shape comparison. We generated the 3D-models of the control embryos and of those with higher neurogenesis (in which Notch-signaling was inhibited), extracted the mesh from them and measured the signed distance between the two mesh surfaces in more than 400,000 points across the models. Then, the measurements were displayed as a color-coded map on the control model. When we compared the rendered hindbrains of both groups of embryos we observed a consistent difference in the overall tissue shape, mainly in the dorsomedial differentiation domain (*Figure 5F–F'*; see color-coded legend showing in blue the highest differences; *Figure 5—video 1*). This result was supported by the previous observation in which the brain ventricle was

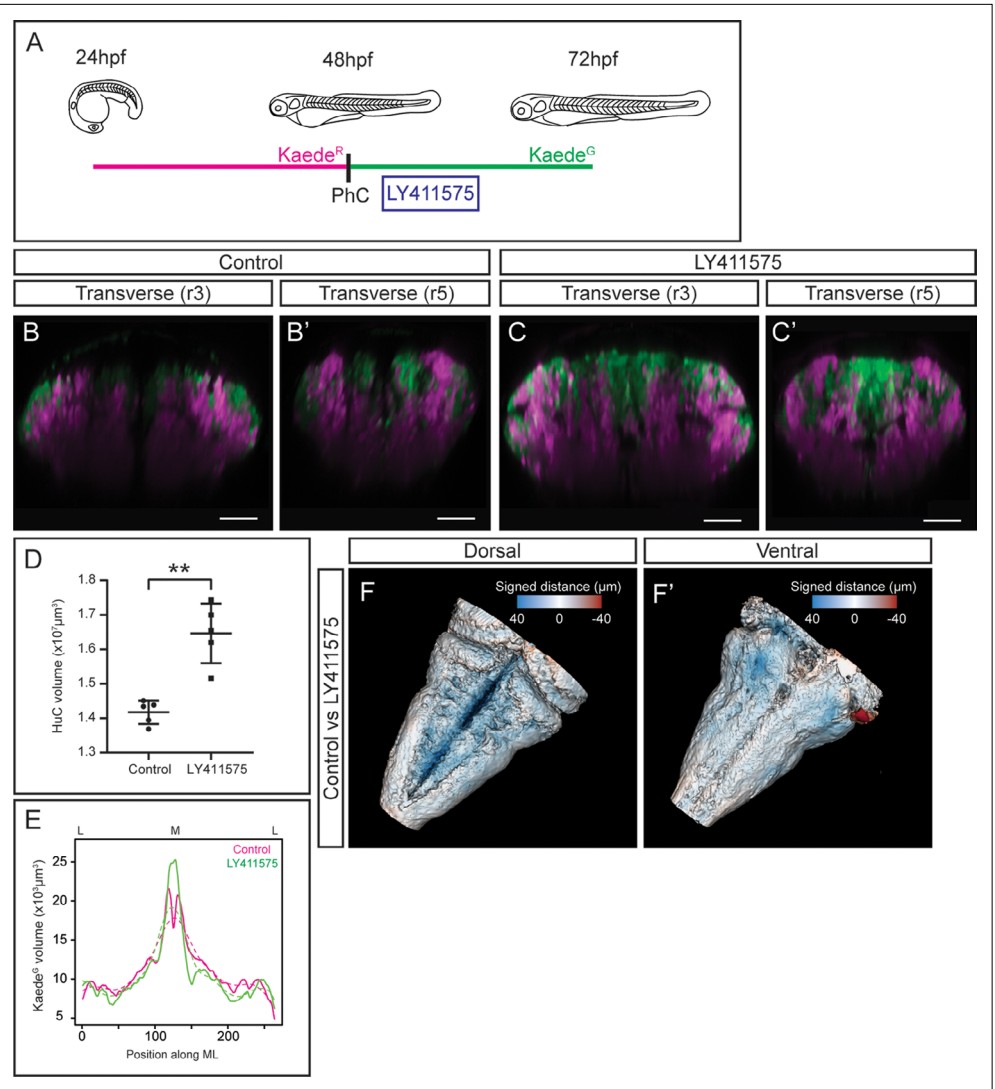

**Figure 5.** Tissue shape analysis to reveal the impact of neurogenesis defects. (**A**) Scheme with the outline of the experiment. HuC:Kaede^Green was photoconverted at 48hpf, embryos were incubated with the gamma-secretase inhibitor LY411575 during 4h, and imaged at 72hpf. As control, HuC:Kaede photoconverted non-treated embryos were used. (**B–C, B'–C'**) Transverse views through r3 and r5 of control (**B–B'**) and LY411575-treated (**C–C'**) embryos. Note the increase of the HuC:Kaede^Green domain upon conditionally inhibiting Notch-signaling, and the consequently remodeling of the HuC:Kaede^Red domain. (**D**) Dot-plot displaying the neuronal differentiation volume in control and LY411575-treated embryos (n=5 embryos each). Welch's two-tailed t-test; $P<0.002$ (**). Note the increase upon Notch-inhibition. (**E**) Quantification of the HuC:Kaede^Green volume in control and treated embryos along the ML axis. Note the accumulation of neurons in the medial differentiation domain of treated embryos. The average value is displayed as solid line and the non-linear regression as dashed line (n=5 embryos each). (**F–F'**) Shape comparison analysis of the neuronal differentiation domain between control and LY411575-treated embryos. Overlay of total 3D-neuronal volumes of control and treated embryos in dorsal (**F**) and ventral (**F'**) views. Color-coded legend indicates the signed distance between the two conditions. Note that the dorsal part is bluer compared with the ventral part, due to the accumulation of more neurons in this domain (n=5 embryos each). Scale bar, 50µm.

The online version of this article includes the following video for figure 5:

**Figure 5—video 1.** Tissue shape comparison after increasing neurogenesis.
https://elifesciences.org/articles/78300/figures#fig5video1

distorted due to the increase of neurons. The increased differentiated domain volume was visible mainly in the dorsomedial part, which was occupied by new neurons and remodeled the neuronal differentiation domain.

## Discussion

One of the main current challenges in biology is the 3D-anatomical reconstruction of the tissues' growth over time together with the cell lineage histories, which requires to extract meaningful biological information from large imaging datasets. Some of the bottlenecks are the required computing power for the visualization and analysis of the data volumes, and the need of user-friendly tools to be paired with imaging techniques. Here, we developed a digital 3D-atlas maker (DAMAKER), which is a dynamic and expandable 3D-tool for monitoring the temporal changes in growth and shape during tissue morphogenesis. We built a digital representation of the developing vertebrate hindbrain from 3D-imaging data collected at different times, which can be used for further quantitative analysis and multilevel modeling. Our goal is to share an established protocol using open-access tools and algorithms that allow standardized and accessible tissue-wide cell population atlas construction and can be used to create an expandable atlas of any tissue in any organism.

In this work, our interest was to explore how neuronal differentiation and morphogenesis might be intertwined during hindbrain development. For this, we registered experimental whole hindbrain images and mapped neuronal differentiation patterns with a unified spatial representation, using confocal imaging setup and leveraging cross-platform segmentation and registration tools. To improve comparison of various signals across embryos that requires an alignment/registration, which is often done manually and is very time-consuming, we used the Fijiyama plugin in FIJI (*Schindelin et al., 2012*). This allowed a multi-modal automatic rigid registration process and a large range of customization, and it permitted the application of a previously computed transform; namely, an alignment can be computed using one channel of an embryo matching the chosen reference and this can be applied to another channel that shares no features with the reference signal. This enables the use of virtually any signal previously added to the atlas, as reference to align others.

The quantitative evaluation of datasets often involves manual annotation, which is time-consuming, introduces biases and often constitutes the major constraint in an evaluation pipeline. To overcome this problem, we used the Trainable Weka Segmentation 3D (TWS), (*Arganda-Carreras et al., 2017*), which combines the image processing toolkit FIJI with the state-of-the-art machine learning algorithms provided in the latest version of the data mining and machine learning toolkit Waikato Environment for Knowledge Analysis (WEKA) (*Hall et al., 2009*). This tool is modular, allows sophisticated data mining processes to be built from the wide collection of base learning algorithms and tools provided, is open-sourced, and can be easily used as a plugin of FIJI. Users' trained classifiers, which are a set of filters and thresholds determined by the algorithm to best match input from the user, can be used repeatedly to segment each dataset. By keeping our pipeline in the FIJI ecosystem, we aim at providing a user-friendly tool to share data, upgrade methodology and adapt the pipeline to a specific experimental set up (model organism, organ etc), thereby making it accessible to anyone.

The digital 3D-brain atlases are essential for neurobiology because they allow to address new questions that can not be answered by the use of the available 2D-atlases. The addition of time as a key parameter is an asset, since it will permit to follow cells upon time, and therefore cell displacements and tissue growth. For this, we made use of an *in vivo* cell birthdating system that allowed us to monitor intervals of neuronal differentiation (*Caron et al., 2008*). Thus, this temporal registration using data acquired *in vivo* at different developmental stages provides important advantages, as it allowed us to (i) dissect the final location of distinct birthdated cell populations, and (ii) obtain a whole-cell population coverage by analyzing the whole differentiated territory, while maintaining the *in vivo* positioning of cells within the structure. Until now, most of the currently used approaches either lack whole population coverage or are limited by expression timing of the gene or transgene; more importantly, although they permit coverage of the whole cell population, they do not allow the user to foreshadow the birthdate of a given cell population at the time of interest. Here, with the combination of photoconversion experiments as a tool to register temporality of neuronal differentiation and DAMAKER, we provide neuronal birthdating 3D-maps and revealed that the order of neuronal differentiation prefigured the spatial distribution of neurons in the hindbrain, with an inner-outer gradient of temporal differentiation.

Usually, expression of the reporter genes in transgenic lines is delayed as compared to the gene expression onset, unless the reporter has been inserted within the given gene locus. Now, we can specifically knock-in genes within a targeted genome site thanks to the CRISPR-Cas technology, but most of the available —and very valuable—transgenic lines were (and still are) generated with random insertion systems that do not always recapitulate the right onset of gene expression. Thus, our tool can be very useful for predicting neuronal birthdates just by analyzing the position of neurons in the differentiation domain as we showed with glutamatergic and GABAergic neurons. Besides that, this developed digital platform will be useful to interrogate phenotypes in neurological diseases, and zebrafish avatars from human neurological disorders could be analyzed in a very simple manner. As a proof-of-concept, we show tissue shape comparisons after modulating neurogenesis using pharmacological treatments. Thus, the combination of several features can provide computational models allowing to merge information from several high-throughput experiments to enlarge our reductionist view, and implement these digital tools for neuronal disorder studies.

In addition, brain morphogenesis studies could use our digital atlas-maker tool, which maps neuronal differentiation domains in the embryonic brain over time and can be easily shared. In the future, DAMAKER can be further improved by using batch processing algorithms (e.g., to assess the effects of pharmacological treatments or gene disruptions) or by adjusting for more geometrically complex tissues or for tissues with less alignment features (such as heart). Last but not least, this pipeline could be applied to time-lapse videos by repeating the process for each frame. This approach will help us to fill the knowledge gap between gene regulatory networks, cell populations and tissue architecture, as it will allow to follow the dynamic events that pattern the early embryonic organs as they happen within the intact system.

## Materials and methods
### Ethics declarations and approval for animal experiments
All procedures were approved by the institutional animal care and use ethic committee (Comitè Etica en Experimentació Animal, PRBB) and the Generalitat of Catalonia (Departament de Territori i Sostenibilitat), and implemented according to National and European regulations. Government and University veterinary-inspectors examine the animal facilities and procedures to endure that animal regulations are correctly followed. The PRBB animal house holds the AAALAC International approval B9900073. All the members entering the animal house have to hold the international FELASA accreditation. The Project License covering the proposed work (Ref 10642, GC) pays particular attention to the 3Rs.

### Zebrafish strains
Embryos were obtained by mating of adult fish using standard methods. All zebrafish strains were maintained individually as inbred lines. Mü4127 transgenic line was used as landmark of rhombomeres 3 and 5; it is an enhancer trap line in which the KalTA4-UAS-mCherry cassette was inserted into the 1.5Kb region downstream of *egr2a/krx20* gene (*Distel et al., 2009*). The Tg[neurog1:DsRed] (*Drerup and Nechiporuk, 2013*) labels neuronal committed cells and the *neurog1*-derivatives. Tg[isl1:GFP] line labels motoneurons (*Higashijima et al., 2000*). Tg[HuC:GFP] (*Park et al., 2000*) and Tg[HuC:Kaede] (*Harrison et al., 2014*) lines were used to label the whole neuronal differentiated domain. To label GABAergic and glutamatergic neurons, Tg[gad1b:lox-DsRed-lox-EGFP] and Tg[vglut2a:lox-DsRed-lox-EGFP] transgenic lines (*Satou et al., 2013*) (denoted Tg[gadb1:DsRed] and Tg[vglut2a:DsRed], respectively) were used.

### Contact for reagent and resource sharing
Further information and requests for resources and reagents should be directed to Cristina Pujades ( cristina.pujades@upf.edu).

### Confocal imaging samples
Zebrafish embryos were treated with phenylthiourea (PTU) at 24hpf, prior to being anesthetized with tricaine and mounted dorsally in 0.9% low melting point agarose on glass-bottom Petri dishes (Mattek) at the desired time. Images were acquired with a Leica SP8 system using PMT or HyD3 detectors

and a 20×glycerol immersion objective, HCX PL APO Lambda blue 20×/0.7 argon laser 30%, Diode 405 nm (DPSS 561; Ready Lasers). *z*-stacks were recorded with 0.57×0.57 × 1.19 µm voxel size. Gain was adjusted to each signal in order to display as little burnt signal as possible.

Stained fixed samples were imaged on a Leica SP8 inverted confocal microscope with 20×glycerol immersion objective and hybrid detectors. For *xyz* confocal cross-sections, *z*-stacks were acquired with a 1.194 µm *z* distance.

## Photoconversion experiments

HuC:Kaede^Green to HuC:Kaede^Red photoconversion was carried out at the indicated embryonic stages with UV light ($\lambda$ =405 nm) using a 20×objective in a Leica SP8 system. Upon exposure to UV light, Kaede protein irreversibly shifts emission from green to red fluorescence ($\lambda$ =516–581 nm). Proper photoconversion was monitored by the appearance of strong Kaede^Red signal under excitation with a 543 nm laser, and the disappearance of Kaede^Green. After photoconversion, embryos were either imaged or returned to embryo medium with PTU in a 28.5 °C incubator to let them grow until the desired stage.

## Whole-mount *in situ* hybridization

Embryo whole mount *in situ* hybridization was adapted from *Thisse and Thisse, 2008*. The *gad1b* riboprobe (*Zhang et al., 2020*) encompassing the nucleotides 1104–2334 of accession number AB183390 was used. The *gad1b* template was generated by PCR amplification by adding the T7 promoter sequence (indicated by capital letters) to the reverse primer (*gad1b* Fw 5'— gat ggt tgc gcg gta taa —3'; *gad1b* Rev 5'— ata tta ata cga ctc act ata gCT TCG TTA AAA GGG TGC —3'). The *vglut* probe was as described in *Higashijima et al., 2004*. For fluorescent *in situ* hybridization, the DIG-labeled probe was detected with TSA-Cy3.

## *In toto* embryo immunostaining

For immunostaining, 4% PFA-fixed embryos were permeabilized with 10mg/ml proteinase K during different times according to the developmental stage (in this case 24–28hpf, 5min), blocked in 2% BSA, 5% goat serum, 0.1% Tween20 in PBS for 3hr at room temperature, and then incubated overnight at 4°C with the primary antibody. The primary antibody used was mouse anti-HuC (1:400). After extensive washing with PBST, embryos were incubated overnight at 4°C with secondary antibody conjugated with Alexa Fluor 633 (1:500).

## EdU experiments

Tg[HuC:Kaede] sibling embryos were photoconverted at 24hpf, and after 6 hr acclimatation the embryos were placed in embryo medium supplemented with 400 µM EdU (Click-iT EdU Imaging Kit, ThermoFisher Scientific), 2% DMSO and PTU. The embryos were immediately placed on ice for 30 min, and later incubated at 28.5 °C for 5 hr 30 min. Then, embryos were rinsed twice in embryo medium with PTU and grown until 48hpf stage. Embryos were fixed in 4%PFA for 4 hr on a shaker, washed, and treated with proteinase K (1:1000 in PBST) for 18 min. After washing them in PBST, they were fixed in 4%PFA for 30 min, washed and revealed with the Click-iT EdU Imaging Kit following manufacturer's protocol. Finally, to recover protein fluorescence, embryos were washed in PBS with 2%BSA and 40 mM EDTA overnight at 4 °C on shake (*Bourge et al., 2015*). The next day they were mounted for imaging.

## Pharmacological treatments

Tg[HuC:GFP] embryos were treated with 10µM of the gamma-secretase inhibitor LY411575 (Sigma-Aldrich) and Tg[HuC:GFP] non-treated were used as control. The treatment was applied into the fish water at 48hpf during 4hr at 28.5°C. Then, embryo medium was changed and they were let to develop until 72hpf, mounted and *in vivo* imaged under a Leica SP8 confocal microscope.

## Image processing and analyses
### Registration algorithm
Fijiyama registration FIJI plugin (*Fernandez and Moisy, 2020*), a registration tool for 3D multimodal time-lapse imaging, was used with four steps of automatic rigid registration for each alignment of

raw samples: 2 steps without and 2 steps with subpixel accuracy; oversized images subsampling was left on. HuC-channel of all samples was registered to a selected HuC signal from a Tg[vglut2a:DsRed;HuC:GFP] 72hpf embryo. Computed transform was then applied to the signals of interest using the 'apply a computed transform to another image' option of the Fijiyama plugin.

### Segmentation algorithm

The Trainable Weka Segmentation FIJI plugin (*Arganda-Carreras et al., 2017*), a machine learning tool for microscopy pixel classification, was used for image segmentation. Training features for the algorithm were kept to mean and variance in order to limit computing power usage, ensuring accessibility. First, classifiers from the very same embryo were generated from supervised training of the algorithm by different users. The users' generated classifiers were used to segment each sample and to obtain the users' average 3D-model from a single embryo. This process was repeated for each embryo (n=5 embryos per sample). Training took about 4 min per step with 3–4 steps to complete, and generation of a mask through a single classifier took about 10 min on a consumer-grade workstation with an 3700 X (8 cores, 16 threads, 3.6 Ghz) AMD processor and RAM 32 GB.

### Processing

Users' and embryos' averaging, as well as consensus thresholds, were performed in FIJI. Homemade batch-processing macros were used to process the whole datasets at once. Macros for batch processing of each of the steps were kept independent to ensure easy adaptability, such as variation in sample size or selected thresholds. Signal consensus among all users was used for both quantifications and models. Signal consensus among 3 or more embryos was used for the generation of the models. Intersection of 3D-models was performed using the image calculator option in FIJI.

### Tissue volume quantification

Embryos' 3D-models were registered together and resliced along different axes. Slices were then quantified using FIJI's 3D object counter. The slicing and quantification of individual slices was automatized with a simple homemade FIJI macro. Quantification along axes was processed through R. We used ratios of signal of interest vs. total differentiation domain volumes of individual embryos to generate a nonparametric regression curve with local linear kernel estimation method (Fan, 2018). The R (R core team 2020) package npregfast version 1.5.1 (*Sestelo et al., 2017*) and the Epanechnikov kernel (with bandwidth parameter selected by cross-validation) were used.

Quantification of 3D-volumes after the application of the pipeline were plotted using either GraphPad Prism 9.0 for global quantifications or R for the axes distributions.

### Tissue shape analysis

Models generated through the pipeline were exported as.OBJ mesh and fed into a python script leveraging the VEDO library (https://zenodo.org/record/6529705#.YtqYZ-xBxgF) to compute the signed distance between the two mesh surfaces. The computed measurements were then displayed as a color-coded map on the reference mesh and histogram of measurement counts was generated.

## Data availability

All datasets supporting our work are included in the manuscript and supported files. z-stacks resulting from the pipeline, and subsequent intersections can be found in the 3D-model collection (Supporting ZIP document). The codes can be found in GitHub https://github.com/cristinapujades/Blanc-et-al.-2022, (copy archived at swh:1:rev:82c8199b774af1b4ff68d7c0ccd2cb8d4197df56; *Blanc, 2022*) with a short usuer manual https://github.com/cristinapujades/Blanc-et-al.-2022/wiki. For further information you may contact Matthias Blanc (matthias.blanc@upf.edu) or Cristina Pujades (cristina.pujades@upf.edu).

## Acknowledgements

The authors thank members from the Pujades lab for help in the training of the algorithms for Trainable Weka Segmentation and for critical insights, and specially Lydvina Meister for help in the in situ hybridization experiments, and Laura Campamà for assistance in final optimization of macros and

writing the user's manual. We like to thank James Sharpe for his advice in the analysis. Fluorescence microscopy was performed at the Advanced Light Microscopy Unit at the CRG, Barcelona. We wish to thank Dr S Higashijima (National Institute for Basic Biology, Japan) and Dr. G Sumbre (IBENS, Paris) for kindly providing the transgenic lines Tg[gad1b:DsRed] and Tg[vglut2a:DsRed].

This work was funded by grants PGC2018-095663-B-I00 and RED2018-102553-T to CP and PGC2018-101643-B-I00 to FU, from Spanish MCIN/AEI (DOI: 10.13039/501100011033) and Fondo Europeo de Desarrollo Regional (FEDER). Department of Medicine and Life Sciences UPF is a Unidad de Excelencia María de Maeztu funded by the Spanish MCIN/AEI (DOI: 10.13039/501100011033), Ref CEX2018-000792-M. CP is a recipient of ICREA Academia award (Generalitat de Catalunya).

## Additional information

### Funding

| Funder | Grant reference number | Author |
| --- | --- | --- |
| Agencia Estatal de Investigación | PGC2018-095663-B-I00 | Cristina Pujades |
| Agencia Estatal de Investigación | PGC2018-101643-B-I00 | Frederic Udina |
| Agencia Estatal de Investigación | CEX2018-000792-M | Matthias Blanc Cristina Pujades |
| Institució Catalana de Recerca i Estudis Avançats | | Cristina Pujades |

The funders had no role in study design, data collection and interpretation, or the decision to submit the work for publication.

### Author contributions

Matthias Blanc, Conceptualization, Resources, Data curation, Software, Formal analysis, Validation, Investigation, Visualization, Methodology, Writing - original draft, Writing - review and editing; Giovanni Dalmasso, Frederic Udina, Software; Cristina Pujades, Conceptualization, Formal analysis, Supervision, Funding acquisition, Investigation, Writing - original draft, Writing - review and editing

### Author ORCIDs

Matthias Blanc http://orcid.org/0000-0002-4892-8161
Frederic Udina http://orcid.org/0000-0002-7155-4153
Cristina Pujades http://orcid.org/0000-0001-6423-7451

### Ethics

Zebrafish (Dario rerio) were treated according to the Spanish/European regulations for the handling of animals in research. All protocols were approved by the Institutional Animal Care and Use Ethic Committee (Comitè; Etica en Experimentació ; Animal, PRBB) and the Generalitat of Catalonia (Departament de Territori i Sostenibilitat), and they were implemented according to European regulations. The Project Licenses covering the proposed work (Ref 10642, Ref 10477, GC) pay particular attention to the 3Rs (Replacement, Reduction, Refinement).

### Decision letter and Author response

Decision letter https://doi.org/10.7554/eLife.78300.sa1
Author response https://doi.org/10.7554/eLife.78300.sa2

## Additional files

### Supplementary files

• MDAR checklist

• Source data 1. z-stacks resulting from the pipeline, and subsequent intersections can be found in the 3D-model collection.

## Data availability

All data generated or analysed during this study are included in the manuscript or in the supplementary files. All code is available in https://github.com/cristinapujades/Blanc-et-al.-2022, (copy archived at swh:1:rev:82c8199b774af1b4ff68d7c0ccd2cb8d4197df56).

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

# Appendix 1

## Appendix 1—key resources table

| Reagent type (species) or resource | Designation | Source or reference | Identifiers | Additional information |
| --- | --- | --- | --- | --- |
| Genetic reagent (*Danio rerio*) | Tg[HuC:GFP] | PMID:11071755 | Tg[HuC:GFP] | |
| Genetic reagent (*Danio rerio*) | Tg[HuC:KAEDE] | PMID:25264359 | Tg[HuC:KAEDE] | |
| Genetic reagent (*Danio rerio*) | Tg[gad1b:DsRed] | PMID:23946442 | Tg[gad1b:lox-DsRed-lox-EGFP] | |
| Genetic reagent (*Danio rerio*) | Tg[vglut2a:DsRed] | PMID:23946442 | Tg[vglut2a:lox-DsRed-lox-EGFP] | |
| Genetic reagent (*Danio rerio*) | Mü4127 | PMID:19628697 | | |
| Genetic reagent (*Danio rerio*) | Tg[neurog1:DsRed] | PMID:23468645 | | |
| Genetic reagent (*Danio rerio*) | Tg[Isl1:GFP] | PMID:10627598 | | |
| Sequence-based reagent (primers) | *gad1b* Fw | PMID:32443726 | | |
| Sequence-based reagent (primers) | *gad1b* Rev | PMID:32443726 | | |
| Sequence-based reagent | *vglut2b* probe | PMID:15515025 | | |
| Antibody | mouse anti-HuC | ThermoFisher | A-21271 | (1:400) |
| Antibody | F(ab')2-Goat anti-Mouse IgG (H+L) Cross-Adsorbed Secondary Antibody, Alexa Fluor 633 | Invitrogen | A21053 | (1:500) |
| Chemical compounds, drug | LY411575 | Sigma-Aldrich | SML0506-5MG | γ-secretase inhibitor |
| Chemical compounds, drug | Click-iT EdU Imaging Kit | ThermoFisher Scientific | C10086 | |
| Software, algorithm | FIJI | PMID:22743772 | | |
| Software, algorithm | Fijiyama registration FIJI plugin | PMID:32997734 | | |
| Software, algorithm | Trainable Weka Segmentation FIJI plugin | PMID:28369169 | TWS | |
| Software, algorithm | VEDO | doi:10.5281/zenodo.4287635 | | |

