## [Editor Report]

This methodological manuscript is of interest to the fields of neural development, tissue morphogenesis, and image analysis technologies. The authors developed an image registration tool and created a digital atlas to reflect the anatomical distribution of neuronal birthdates in the developing zebrafish hindbrain. The provided resources can be very useful to monitor temporal changes in tissue growth.

---

## [Decision Letter]

**Decision letter after peer review:**

Thank you for submitting your article "The Digital 3D-Atlas MAKER (DAMAKER): a dynamic and expandable digital 3D-tool for monitoring the temporal changes in tissue growth during hindbrain morphogenesis" for consideration by *eLife*. Your article has been reviewed by 3 peer reviewers, and the evaluation has been overseen by a Reviewing Editor and Richard White as the Senior Editor. The following individual involved in the review of your submission has agreed to reveal their identity: Venera Weinhardt (Reviewer #3).

Essential revisions:

All reviewers agreed that the paper needs to improve significantly on the following three points:

1 – To improve the temporal content of the method by additional experiments on time-lapse imaging and corresponding controls.

2 – To show inter-individual variability, which should be possible without additional experiments by incorporation of (most probably) existing data.

The details on points 1 and 2 are outlined in the constructive recommendations to the authors and public reviews. Please also note that the reviewers consented that while analyses of mutants would be nice to add, they are not required for the publication of this article.

3 – To deposit software along with explanations/manuals on how to use and extend it further. The two links that are currently provided send to the identical webpage on which neither Macros nor R code can be found.

*Reviewer #2 (Recommendations for the authors):*

1. Much higher 3D magnifications are needed in order to evaluate the morphogenetic changes of particular cell types and domains in a greater resolution.

2. More experimental data is required in order to further convince how the new imaging protocols provide many more detailed answers compared to traditional imaging tools.

3. Validation of the technology's strength should be provided by asking a physiological question about the developing hindbrain. Many relevant mutant lines are available (including in the author's lab) that can serve to demonstrate how the DAMAKER protocol is indeed an innovative and original strategy to study brain development in a manner that was absent in previous studies.

4. As opposed to the numerous notions of the authors regarding the development of a novel code to provide a temporal 3D atlas, the actual data presented to demonstrate this capacity is rather limited. Hence, the text should be more balanced in that aspect.

*Reviewer #3 (Recommendations for the authors):*

The manuscript is readable but at places is difficult to understand due to a lack of information and broader terms. The following aspects can be improved:

– It is stated that manual segmentation of masks was performed. Masks of what exactly? The fluorescence imaging is per se is either signal (1) or not (0). Therefore, the whole purpose of masking is therefore unclear.

– Supplementary figure 1, panels E and F please make sure to clarify what is measured and what is calculated value? As I understand NDD is measured and PD is simply calculated by substruction, thus the same std and inverted look of plots.

– A supplementary figure with corresponding stages of zebrafish development would be very helpful for those outside of the zebrafish community.

– I think it is important to show individual variability of the data, which would help to understand the difficulties of the atlas making and justify the development of a pipeline. Please include data on multiple datasets with the same marker.

– Line 206 "Once the alignment of the HuC signal was accomplished,…". How the alignment was verified? Fijiyama to my memory uses an iterative approach, was the number of iterations fixed to 4? Was its success verified and how?

– Throughout the manuscript when reporting on the quantitative analysis of a tissue, in particular volume, you use "neurons are born". That is misleading. The analysis pipeline does not count the number of neurons but rather the volume occupied by a tissue. Please make sure to state it at least in the beginning.

– In all figures and videos please do not use a red-green colour combination. 20% of readers are colorblind.

– Please include the AP and LML coordinate system on the figures to ensure that readers with other backgrounds can understand it as well.

Finally, I could not find supporting software on your web page: https://www.upf.edu/web/pujadeslab/open-science. Considering the focus of your manuscript this is essential. Next time, please ensure the availability of your software during the review process.

[Editors’ note: further revisions were suggested prior to acceptance, as described below.]

Thank you for resubmitting your work entitled "A dynamic and expandable Digital 3D-Atlas MAKER for monitoring the temporal changes in tissue growth during hindbrain morphogenesis" for further consideration by *eLife*. Your revised article has been evaluated by the three original reviewers, as well as by Richard White (Senior Editor) and a Reviewing Editor.

We much appreciate the work you put into the revised version, and think that it is now almost suitable for publication in *eLife*. There is still one request on the assessment of the inter-individual variability that will be highly useful to implement, also in the perspective of the future usefulness of your work for others.

The authors quantified the interindividual variability more carefully in Figure 1-Supplement 1 and Figure 3 – Supplement 1, with the purpose of demonstrating that "interindividual variability is low". However, in all the grayscale representations, variability is clearly visible in that some non-white volume area have at least 2~3 layers of cells, and even the HuC volume of the fish quantified at the same experimental condition could have a variability of 20% (Fig1S1F and Fig3S1A andB). Instead of only claiming the "low variability", the authors should have a quantitative measurement of this variability in the manuscript, and provide the confidence level of the birthdate map. This will essentially help the users to make a decision on whether an area of interest should be followed up or is beyond the precision that can be measured by the atlas.

---

## [Author Response]

Essential revisions:All reviewers agreed that the paper needs to improve significantly on the following three points:1 – To improve the temporal content of the method by additional experiments on time-lapse imaging and corresponding controls.

In order to improve the temporal content of the method we have performed and included the following experiments:

1.1 To show that there is a progressive temporal growth of the HuC-domain, we have included an additional temporal window of neuronal differentiation and provide the neuronal birthdating 3D-maps at 48hpf and 72pf (new Figure 3; Figure 3– videos 1–2). We have quantified the contribution of the distinct birthdated neuronal populations to the HuC-domain (new Figure 3—figure supplement 2).

1.2 In the same line, to show the progressive temporal growth of the glutamatergic and GABAergic neuronal domains within the HuC we provide an additional temporal window for these cell populations (new Figure 4; Figure 4– video 1). We have quantified the contribution of the neurons born at different times to the corresponding vglt2a/gad1b-domains and compared their growth dynamics (Figure 4E–G).

1.3 To further support the observation that indeed newly born cells were orderly incorporating into the neuronal differentiation domain, we combined HuC:Kaede^Green^ photoconversion with an additional age-labeling method such as EdU. We photoconverted HuC:Kaede^Green^ at 24hpf, pulsed embryos at 30hpf with EdU to label proliferating cells, and let them grow until 48hpf. We could observe that new neurons born after 24hpf were found by 48hpf in the newly generated neuronal differentiation domain (Figure 3D, E—E’). This responds to point 3 of Reviewer 1 asking to show the new cells incorporated between the two time points are reflected in the growing unphotoconverted population.

1.4 To reveal that the HuC:Kaede^Red^ cells labeled in an early conversion are strictly included in the HuC:Kaede^Red^ domain labeled in a late conversion, we overlaid the differentiation domains contributed at each time interval (Figure 3—figure supplement 1F–H, F’–H’). This responds to point 3 of Reviewer 1.

1.5 To demonstrate that indeed the birthdating 3D-maps could be used as proxy to infer birthdate, we verified that glutamatergic and GABAergic neurons were born before 24hpf, as our tool predicted and included it in new Figure 4. We performed *in situ* hybridizations with *vglut2a* and *gad1b* probes at early stages of embryonic development (24hpf) while monitoring the neuronal differentiation domain. Indeed, glutamatergic and GABAergic neurons were already specified at 24hpf and they were located within the HuC territory (Figure 4H–H’’), although the transgenic lines did not display the reporter expression at this stage.

1.6 As additional controls to demonstrate that different type of experiments can be combined, we evaluated any possible differences within the conditions, through quantification of HuC volumes at 48hpf and 72hpf in embryos in which photoconversion of Hu:Kaede^Green^ was performed at different stages (new Figure 3—figure supplement 1A–B). These additional volume analyses show that overall hindbrain volumes are comparable even after photoconversion, and that embryos are properly/homogenously staged.

2 – To show inter-individual variability, which should be possible without additional experiments by incorporation of (most probably) existing data.

2.1. To evaluate the accuracy of the registration, interindividual variability of different signals was assessed. Dorsal and transverse projection views overlaying five different specimens and displaying the interindividual variability as a gray scale is now shown by comparison of average signal prior to threshold among individuals (new Figure 1—figure supplement 1A–E, A’–E’).

2.2. To show interindividual volume variability, the size of the neuronal differentiation territory (HuC-domain) in different transgenic backgrounds was assessed displaying no significative differences (new Figure 1—figure supplement 1F).

2.3. To demonstrate that birthdating 3D-maps displayed low interindividual variability, we verified that photoconversion of the HuC-:KaedeGreen at different stages was consistent in all the analyzed embryos. For this, we first verified that the overall HuC-volumes did not dramatically change among embryos that underwent HuC:KaedeGreen photoconversion at different stages (new Figure 3—figure supplementary 1A–B). Second, embryos in which HuC:KaedeGreen was photoconverted at different times were imaged at 72hpf, and the HuC:KaedeRed territories of distinct embryos were overlaid and displayed as a gray gradient (new Figure 3—figure supplementary 1C–E, C’–E’).

The details on points 1 and 2 are outlined in the constructive recommendations to the authors and public reviews. Please also note that the reviewers consented that while analyses of mutants would be nice to add, they are not required for the publication of this article.

2.4 We fully agree with the Referees, and although we do not include the analysis of mutants, we have decided to take the newly developed protocol one step further to serve as a proof-of-concept study by using fish with normal or aberrant neurogenesis as Referee 2 commented. For this, we include the comparative analysis of control embryos with those in which the Notch-signaling was downregulated and therefore they have increased neurogenesis. Since it has been reported that hindbrain cells respond to Notch (Hevia, Engel-Pizcueta, Udina, and Pujades, 2022; Nikolaou et al., 2009), we conditionally downregulated Notch-signaling by treating Tg[HuC:GFP] embryos with an inhibitor of the γ-secretase, LY411575, and assessed the changes in the neuronal differentiation domain volume and 3D-shape. Results are displayed in new Figure 5 and Figure 5– video 1. This experiment shows that our pipeline can be of use for the analyses of 3D-shape changes too.

3 – To deposit software along with explanations/manuals on how to use and extend it further. The two links that are currently provided send to the identical webpage on which neither Macros nor R code can be found.

We apologize for not including the right links to obtain the macros and the R code that were used in the manuscript. The used software along with explanations on how to use and extend it further can be found at:

https://github.com/cristinapujades/Blanc-et-al.-2022

We uploaded the updated macros with a short description for better comprehension. In addition, we included a brief description on how to adapt them for other uses.

Reviewer #2 (Recommendations for the authors):1. Much higher 3D magnifications are needed in order to evaluate the morphogenetic changes of particular cell types and domains in a greater resolution.

See point 1 in Reviewer 2.

2. More experimental data is required in order to further convince how the new imaging protocols provide many more detailed answers compared to traditional imaging tools.

See point 1 in Essential Revisions.

3. Validation of the technology's strength should be provided by asking a physiological question about the developing hindbrain. Many relevant mutant lines are available (including in the author's lab) that can serve to demonstrate how the DAMAKER protocol is indeed an innovative and original strategy to study brain development in a manner that was absent in previous studies.

See point 2.4 in Essential Revisions.

4. As opposed to the numerous notions of the authors regarding the development of a novel code to provide a temporal 3D atlas, the actual data presented to demonstrate this capacity is rather limited. Hence, the text should be more balanced in that aspect.

We provide code and strategies to automatize processes and to allow this type of approaches to be feasible in a decent amount of time. We do not provide a new registration or segmentation approach as stated in the text.

Reviewer #3 (Recommendations for the authors):The manuscript is readable but at places is difficult to understand due to a lack of information and broader terms. The following aspects can be improved:

We have improved the writing and included more and better explanations.

– Supplementary figure 1, panels E and F please make sure to clarify what is measured and what is calculated value? As I understand NDD is measured and PD is simply calculated by substruction, thus the same std and inverted look of plots.

See response to point 1 to Reviewer 1.

– A supplementary figure with corresponding stages of zebrafish development would be very helpful for those outside of the zebrafish community.

We followed the Reviewer’s advice, provided schemes of embryos at the different stages of embryonic development covered in the manuscript, and we have better depicted the experiments outline.

– I think it is important to show individual variability of the data, which would help to understand the difficulties of the atlas making and justify the development of a pipeline. Please include data on multiple datasets with the same marker.

See point 2 in Essential Revisions.

– Line 206 "Once the alignment of the HuC signal was accomplished,…". How the alignment was verified? Fijiyama to my memory uses an iterative approach, was the number of iterations fixed to 4? Was its success verified and how?

The number of iterations used was 4 as mentioned in the protocol. Alignment accomplishment can be verified through Fijiyama itself, and we further confirmed it through tissue shape analysis. Since we concluded that the approach was robust enough, we did not consider necessary to discuss it, especially since it is done on Python and we wanted to keep the pipeline as much as possible in FIJI. We have included it in this new version.

– Throughout the manuscript when reporting on the quantitative analysis of a tissue, in particular volume, you use "neurons are born". That is misleading. The analysis pipeline does not count the number of neurons but rather the volume occupied by a tissue. Please make sure to state it at least in the beginning.

Yes, the Reviewer is right and we followed her advice.

– In all figures and videos please do not use a red-green colour combination. 20% of readers are colorblind.

We have changed them.

– Please include the AP and LML coordinate system on the figures to ensure that readers with other backgrounds can understand it as well.

We have either included them or mentioned the orientation in the figure legends.

Finally, I could not find supporting software on your web page: https://www.upf.edu/web/pujadeslab/open-science. Considering the focus of your manuscript this is essential. Next time, please ensure the availability of your software during the review process.

See point 3 in Essential Revisions.

References

Caron, S. J. C., Prober, D., Choy, M., and Schier, A. F. 2008. In vivo birthdating by BAPTISM reveals that trigeminal sensory neuron diversity depends on early neurogenesis. *Development*, 135(19): 3259–3269.

Distel, M., Wullimann, M. F., and Köster, R. W. 2009. Optimized Gal4 genetics for permanent gene expression mapping in zebrafish. *Proceedings of the National Academy of Sciences of the United States of America*, 106(32): 13365–13370.

Fabian, P., Tseng, K.-C., Smeeton, J., Lancman, J. J., Dong, P. D. S., et al. 2020. Lineage analysis reveals an endodermal contribution to the vertebrate pituitary. *Science*, 370(6515): 463–467.

Hevia, C. F., Engel-Pizcueta, C., Udina, F., and Pujades, C. 2022. The neurogenic fate of the hindbrain boundaries relies on Notch3-dependent asymmetric cell divisions. *Cell Reports*, 39(10): 110915.

Kesavan, G., Chekuru, A., Machate, A., and Brand, M. 2017. CRISPR/Cas9-Mediated Zebrafish Knock-in as a Novel Strategy to Study Midbrain-Hindbrain Boundary Development. *Frontiers in Neuroanatomy*, 11: 52.

Kesavan, G., Machate, A., Hans, S., and Brand, M. 2020. Cell-fate plasticity, adhesion and cell sorting complementarily establish a sharp midbrain-hindbrain boundary. *Development*, 147(11). https://doi.org/10.1242/dev.186882.

Nikolaou, N., Watanabe-Asaka, T., Gerety, S., Distel, M., Köster, R. W., et al. 2009. Lunatic fringe promotes the lateral inhibition of neurogenesis. *Development*, 136(15): 2523–2533.

Voltes, A., Hevia, C. F., Engel-Pizcueta, C., Dingare, C., Calzolari, S., et al. 2019. Yap/Taz-TEAD activity links mechanical cues to progenitor cell behavior during zebrafish hindbrain segmentation. *Development*, 146(14). https://doi.org/10.1242/dev.176735.

[Editors’ note: further revisions were suggested prior to acceptance, as described below.]

The authors quantified the interindividual variability more carefully in Figure 1-Supplement 1 and Figure 3 – Supplement 1, with the purpose of demonstrating that "interindividual variability is low". However, in all the grayscale representations, variability is clearly visible in that some non-white volume area have at least 2~3 layers of cells, and even the HuC volume of the fish quantified at the same experimental condition could have a variability of 20% (Fig1S1F and Fig3S1A andB). Instead of only claiming the "low variability", the authors should have a quantitative measurement of this variability in the manuscript, and provide the confidence level of the birthdate map. This will essentially help the users to make a decision on whether an area of interest should be followed up or is beyond the precision that can be measured by the atlas.

We made quantitative measurements of the interindividual variability and included in Figure 1 – Supplement 1G and Figure 3 – Supplement 1B and D. These quantifications indicate that there is an average of maximum of 15% of difference between the HuC volume of the individuals and the HuC volume of the generated model. We included these data in the manuscript (pages 7, 11) and in the corresponding figure legends.

We made changes in the color-code in Figure 3E and switched to colors with more contrast. We specified the difference between H, H' and H'’ in Figure 4H. We tried several color combinations for Figure 4H; however, we did not change it because the current one was the best.

We have updated the wiki for a better comprehension.